(c) Author(s) 2019. CC BY 4.0 License.



# CAFE: a new, improved non-resonant laser-induced fluorescence instrument for airborne in situ measurement of formaldehyde

Jason M. St. Clair[1,2], Andrew K. Swanson[1,3], Steven A. Bailey[1], Thomas F. Hanisco[1]

[1]Atmospheric Chemistry and Dynamics Laboratory, NASA Goddard Space Flight Center, Greenbelt, MD, 20771, USA
[2]Joint Center for Earth Systems Technology, University of Maryland Baltimore County, Baltimore, MD, 21228, USA
[3]Goddard Earth Sciences Technology and Research, Universities Space Research Association, Columbia, MD, 21046, USA

*Correspondence to*: Jason M. St. Clair (jason.m.stclair@nasa.gov)

**Abstract.** NASA Compact Airborne Formadehyde Experiment (CAFE) is a non-resonant laser-induced fluorescence instrument for airborne in situ measurement of formaldehyde (HCHO). The instrument is described here with highlighted improvements from the predecessor instrument, COmpact Formaldehyde FluorescencE Experiment (COFFEE). CAFE uses a 480 mW, 80 kHz laser at 355 nm to excite HCHO and detects the resulting fluorescence in the 420-550 nm range. The fluorescence is acquired at 5 ns resolution for 500 ns and the unique time profile of the HCHO fluorescence provides

measurement selectivity. CAFE achieves a 1σ precision of 160 pptv (1 s) and 90 pptv (10 s) for [HCHO] = 0 pptv. CAFE has successfully flown on multiple aircraft platforms and is particularly well-suited to high altitude research aircraft or small air quality research aircraft where high sensitivity is required but operator interaction and instrument payload is limited.

## 1 Introduction

Formaldehyde (HCHO) in Earth's atmosphere is primarily produced during the oxidation of volatile organic compounds (VOCs) including methane and isoprene, with minor direct emissions from biomass burning and fossil fuel combustion (Fortems-Cheiney et al., 2012). The atmospheric lifetime of HCHO is a few hours against photolysis and reaction with OH. HCHO is thus sensitive to local variability in both VOC emissions/transport and oxidation capacity. HCHO measurements are consequently valuable for deriving continental-scale isoprene emission fluxes (Palmer et al., 2003), remote-atmosphere

maps of column OH (Wolfe et al., 2019), and regional/continental-scale maps of organic aerosol (Liao et al., 2019). Deep convective transport of oxygenated VOCs (OVOCs) such as HCHO (e.g., Fried et al., 2016), or transport of OVOC precursors (e.g., Apel et al., 2012), can account for a significant portion of the upper tropospheric (UT) HO$_x$ budget (e.g., Ren et al., 2008) and thereby impact UT O$_3$ levels (e.g., Crawford et al., 1999). HCHO measurements in the UT are necessary to close the HO$_x$ budget (Jaeglé et al., 2001) and can also be used in the UT and lower stratosphere (LS) as a

qualitative tracer of recent convective activity. Monitoring HCHO is also important for public health: the EPA classifies



HCHO as a hazardous air pollutant (HAP) and it is estimated to pose the highest cancer risk of all the HAP species (Strum and Scheffe, 2016).

A number of measurement techniques have been applied to in situ atmospheric measurements of HCHO and are reviewed elsewhere (Fried et al., 2008; Hak et al., 2005; Kaiser et al., 2014; Shutter et al., 2019). The most common techniques are absorption spectroscopy (e.g., Richter et al., 2015; Washenfelder et al., 2016), wet chemistry (e.g., Aiello and McLaren, 2009), and laser-induced fluorescence (LIF) (e.g., Cazorla et al., 2015; Hottle et al., 2009). The different techniques offer trade-offs between cost, instrument size, sampling time resolution, and sensitivity. A new approach, non-resonant LIF (NR-LIF), was recently developed to provide a more compact, lower cost instrument with simpler operation than typical HCHO LIF instruments (St. Clair et al., 2017). The NR-LIF approach trades the wavelength-tunable excitation laser traditionally used in HCHO LIF for a fixed wavelength 355 nm laser that excites multiple HCHO absorption features. The broad, fixed wavelength laser cannot tune on and off a single absorption feature to uniquely detect HCHO. Measurement selectivity is provided by carefully selected fluorescence optical filters and rapid data acquisition that enables resolving the time profile of the HCHO fluorescence signal.

The NR-LIF HCHO instrument described in St. Clair et al., 2017, Compact Formaldehyde FluorescencE Experiment (COFFEE), was specifically designed for operation on the Alpha Jet stationed at NASA Ames Research Center and is not easily integrated onto other aircraft platforms. The Compact Airborne Formaldehyde Experiment (CAFE) instrument described here uses the same general measurement approach as COFFEE but with updated components and design changes for operation under the demanding conditions of the NASA ER-2 aircraft. The compact instrument design combined with reliable, commercial subsystems makes CAFE well-suited for scientific projects on high altitude aircraft, where high sensitivity is required but size and weight are limited, as well as tropospheric aircraft platforms with limited payload. In this paper we present the instrumental details of CAFE, including modifications from the original COFFEE design that stem from an improved understanding of the HCHO NR-LIF technique.

## 2 Instrument description

The CAFE instrument is enclosed by a 42 cm x 53 cm x 30 cm (L x W x H) chassis (Fig. S1) with all plumbing and electrical connections on one side of the instrument. The main subsystems of the instrument are shown in Fig. 1 with the chassis lid removed and are described in more detail in the following sections.

### 2.1 Laser

The light source in CAFE is a Spectra-Physics Explorer One 355-1W (Fig. 1, item A) that provides 480 mW of pulsed 355 nm light. The laser is operated at 80 kHz (adjustable 0-300 kHz) to reduce the peak power and minimize risk of mirror



damage. Pulses are ~10 ns wide (FWHM) and the laser bandwidth is approximately 1 nm. Laser control is performed by a RIO data acquisition system (Section 2.5) through RS232 communication.

## 2.2 Optical system

Figure 2 illustrates the details of the CAFE optical plate (Fig. 1, item B). The laser beam is directed into the optical cell by
three antireflection (AR)-coated dielectric (Acton Optics) steering mirrors (Fig. 2, item A) with a collimating lens (F=100 mm) and half wave plate (AR coated, Optisource) in between mirrors 2 and 3 (together Fig. 2, item B). AR-coated fused silica (CVI) cell windows (Fig. 2, item C) provide the vacuum seal for the optical entrance and exit of the optical cell. Optical baffles (Fig. 2, item D) along the entrance and exit 'baffle arms' of the optical cell significantly reduce the stray light reaching the detection volume. Baffle opening diameter increases from entrance to exit over the 7 baffles with three sizes:
2.5 mm, 3 mm, 3.5 mm. The baffles closest to the detection volume are designed in-house and commercially coated (Acktar, vacuum black) while the other 5 baffles are commercially available spatial filter apertures (National Aperture). Cylindrical spacers between the baffles are coated (Acktar, magic black) though background counts are minimally sensitive to the spacer surface for spacers not adjacent to the detection volume. The inner surface of the detection cell is black anodized to reduce stray light. To achieve a more compact optical layout, the ends of the baffle arms are at 90° to the rest of the arm. The 90°
turns are provided by two fixed steering mirrors (Fig. 2, item A). After exiting the optical cell, the laser beam passes through a beam sampler (Thorlabs) (Fig. 2, item E) which directs a few percent of the beam to a power meter (Fig. 2, item F) comprised of a ground glass diffuser (Thorlabs DGUV 10-600), an absorption filter (Thorlabs FGUV11), and an amplified photodiode (OSI 555-UV). The majority of the laser beam passes through the beam sampler and into a beam dump.

Sample air flows through the detection cell perpendicular to the laser beam path and detection axes. Aspheric lenses (NA=0.66, AR-coated, Edmund Optics) on opposite sides of the detection cell (Fig. 2, item G) image the intersection of the sample air flow and the laser path to collect the HCHO fluorescence (~355-550 nm). For each detection arm, the collected light is turned 90° by a mirror (right angle prism dielectric, Thorlabs) (Fig. 2, item H) and directed toward the optical filters (Fig. 2, item I), a lens (F=75 mm, AR-coated, Thorlabs) (Fig. 2, item J), and the photomultiplier tube (PMT) (Hamamatsu
H7360-02 MOD) (Fig. 2, item K). The optical filters for the two detection axes differ from each other and from the filters described in St. Clair et al., 2017. The filters were selected to transmit the maximum amount of HCHO fluorescence while blocking the laser wavelength and Raman scatter from $N_2$, $O_2$, and $H_2O$, as shown for axis 1 in Fig. 3a and axis 2 in Fig. 3b. The primary motivation for the change in filters from St. Clair et al., 2017 was the realization that $H_2O$ Raman signal, at high $[H_2O]$, adversely affected data processing as discussed in the Supplement (Fig. S2). Replacing some of the long pass filters
with red-shifted alternatives reduces the instrument sensitivity but improves accuracy at higher ambient water vapor. Axis 1, proceeding toward the PMT, uses an AR-coated 370 nm long pass absorption filter (Hoya Candeo Optronics), a 405 nm long pass interference filter (Semrock), and a 420 nm long pass absorption filter (Thorlabs). Axis 2, proceeding toward the PMT, has an AR-coated 370 nm long pass absorption filter (Hoya Candeo Optronics), a custom 11-band band pass interference



filter (Semrock), a 405 nm long pass interference filter (Semrock), and a 420 nm long pass absorption filter (Thorlabs). As of April 2018, COFFEE also uses these fluorescence filters.

### 2.3 Air sampling

The details of the inlet and sample tubing external to the instrument are dependent on the aircraft platform for a given experiment. CAFE has used previously existing inlets for various aircraft platforms, e.g. the inlet described in Cazorla et al., 2015 for the NASA DC-8. A new inlet was designed in 2016 for use on the high-altitude NASA ER-2 aircraft. The inlet body consists of a nacelle mounted to the end of an air foil that attaches to the ER-2 superpod midbody via an adapter plate (Fig. 4). The center of the nacelle is held 10.1 cm from the adapter plate to extend sampling beyond the boundary layer of

the superpod. Ram pressure is generated from the tapered shape of the nacelle and allows a higher instrument operating pressure than possible with ambient pressure sampling. Sample air is pulled into the instrument through a heated 6.35 mm OD Silcosteel tube that protrudes into the nacelle perpendicular to the ambient air flow. The tube is chamfered 15° away from the direction of flight to reject particles larger than $\sim 1$ μm. 28 W of heat is provided by a cartridge heater (SunRod) mounted in a copper block that clamps around the sample tube, and is controlled to 30° C by a proportional controller

(Minco CT335) with protection from thermal run-away using a bimetallic hermetic thermostat.

Inside the pod, 1.3 m of tubing (fluorinated ethylene propylene (FEP) or stainless steel) carries the sample flow from the inlet to the instrument. A particle filter (Balston 9922-05-DQ) in-line with the tubing mitigates aerosol-related artifacts (St. Clair et al., 2017). Sample air enters CAFE through a Swagelok VCO bulkhead fitting and 7 cm of FEP tubing before

reaching a pressure controller (Fig. 1, item C). The pressure controller consists of a custom Ni-plated valve block and an actuator (iQ Valve) with MKS control electronics. 22 cm of FEP tubing connects the pressure controller to the detection cell. The final section of tubing is covered in an opaque black shrink wrap to reduce background counts from ambient light while the chassis cover is off for alignment. All metal surfaces before the detection cell are coated with a fluorocarbon (FluoroPel, Cytonix) to reduce HCHO surface interactions and improve instrument time response. For tropospheric

measurements, detection cell pressure is maintained at 10.7 kPa by the pressure controller and mass flow through the instrument is typically ~2.3 slm. For stratospheric measurements, the cell pressure is necessarily reduced 4 kPa to maintain control at high altitudes. Air passes through the detection cell and along the bottom of the optical plate where it proceeds to the pump (Vacuubrand MD-1, Fig. 1 item D), first passing through a ~70 cm³ ballast volume that muffles the pressure fluctuations from the pump. A fraction of the airflow leaves the detection cell via the baffle arms to flush that volume and

rejoins the main flow after the ballast volume. Pump exhaust is plumbed from the instrument to the inlet where it is directed out the back of the air foil (Fig. 4).




## 2.4 Thermal management

Temperature control, to varying degrees, is necessary for some of the instrument subsystems. Adequate heat removal is required for proper laser performance: two thermoelectric cooler (TEC) elements (TE Technology) provide unidirectional thermal control to cool the laser mounting plate and maintain temperature to ≤ 303 K. TECs are thermally attached to the

optical plate with additional heat dissipation provided by heat sinks mounted to the underside of the optical plate. Air is drawn up through the bottom of the instrument chassis by two fans (22 L s⁻¹ each) that blow directly onto the heat sinks. The fans are controlled by a 30°/35° C off/on controller (NuWave Technologies) to avoid cooling at low temperatures. The pump is a significant heat source and so the four fans that exhaust air out of the chassis (visible in Fig. S1) pull air across the pump to prevent its heat from adversely affecting the laser and other subsystems. The exhaust fans are also controlled to

turn off below 30° C.

The pump operates best when the ambient temperature is 10-40° C. To maintain pump function in the unheated pods of high-altitude aircraft, 31 W of cartridge heaters are installed on the pump. Cartridge heaters are also used to heat the pressure controller (24 W) and detection cell (24 W) to 30° C. Polyimide foil heaters provide 70 W of heat to the optical plate, also

maintained at 30° C. All heaters are controlled by proportional controllers (Minco CT335) and protected from thermal run-away using thermostats.

## 2.5 Data acquisition

CAFE utilizes a National Instruments CompactRIO (RIO) data acquisition system (Fig. 1, item E) that runs a real-time operating system. Signal from each PMT is counted with 5 ns resolution by two channels of a NI 9402 high speed digital

I/O module, which is triggered by a TTL pulse from the laser. The signals from the PMTs are delayed by 50 ns with a passive delay circuit (Data Delay Devices, 1515 series) so that the PMT signals arrive after the laser TTL pulse. The digital I/O module, along with analog NI 9205 (input) and NI 9264 (output) modules, interfaces with a main processor module and a backplane with a field programmable gate array (FPGA). Data are recorded at 10 Hz for non-gated (continuous) and time-gated counts, and at 1 Hz for all other data including instrument diagnostics and time-resolved counts. 10 Hz data are used

mainly as an alignment metric. The 1 Hz time-resolved count data consists of a 500 ns time profile of counts, resolved to 100 discrete 5 ns bins, immediately following the laser TTL pulse. HCHO mixing ratio data is generated from the 1 Hz time-resolved count data.

## 2.6 Data processing

The data processing approach for CAFE is almost identical to the methods described in St. Clair et al., 2017 for COFFEE

with minor changes related to laser pulse timing unique to each instrument. Briefly, data from each detection axis is processed by performing three sequential fits to the 1 Hz, 5 ns binned data (0-500 ns from trigger). The first fit is a single



parameter fit to remove the slowly decaying background 'long-lived' signal from the time profile and give a zero baseline at high bin number where HCHO does not contribute to the signal. After the long-lived signal is removed, the 0-500 ns time profile is fit using a two-parameter nonlinear least squares optimization as shown in Fig. 5. The two parameters of the fit are scaling factors to the profiles that represent the HCHO (dashed red) and non-HCHO 'air' (dashed blue) portions of the time

profile, which are determined by laboratory calibration and a pre-flight zero run. While the representative profiles, or exemplars, span the full 500 ns acquisition period, the fit (black line) is performed only from bin 11 to bin 56 (50-280 ns) in a trade-off between HCHO data precision and fit quality. The third step is a single parameter nonlinear least squares fit with only the HCHO portion of the fit varying and the non-HCHO portion fixed to a smoothed version of the two-parameter fit output, improving the precision of the HCHO data compared to the output of the two-parameter fit. The calibration scalar is

then applied to the single parameter fit output to yield HCHO in ppbv.

Data can also be processed using delayed-gate counts at 1 Hz or 10 Hz to give higher precision HCHO data (see Sect. 3.2) that is potentially more susceptible to measurement artifacts. Delayed-gate count data sums the PMT counts over the same time window as the exemplar fit (bin 11 to bin 56, 50-280 ns), keeping the background signal low by excluding the prompt

signal from scatter. HCHO data in ppbv is generated by subtracting the background counts determined from a preflight zero run and applying the calibration factor. More details on the fits, the 'exemplar' time profiles, and processing with gated counts are available in St. Clair et al., 2017.

### 3 Performance

### 3.1 Calibration and sensitivity

CAFE calibration is performed in the same manner as for the ISAF and COFFEE instruments. Flow (0-50 standard $cm^3$ $min^{-1}$, sccm) from a ~500 ppbv HCHO in $N_2$ gas cylinder is added to a main flow (~4 standard L $min^{-1}$, sLm) of ultrahigh purity air to give HCHO typically ~ 0-10 ppbv. The ~4 sLm flow is greater than the instrument sample flow (~2.3 sLm) to speed up the time response of the calibration set-up, with the excess flow exhausting to the room through a 'T' at the instrument chassis. The HCHO mixing ratio is stepped through 3-5 different flows with >45 minutes at each level to provide

confidence that the calibration set-up has equilibrated. The mixing ratio of the HCHO calibration cylinders are verified before and after each aircraft mission using IR absorption as described in Cazorla et al., 2015. Uncertainty in the gas standard combines with uncertainty in the calibration system flow measurements to give a calibration uncertainty of ±10%. For the purposes of comparison with other LIF instruments, the ungated count sensitivities are 1.3 counts $s^{-1}$ $mW^{-1}$ $ppbv^{-1}$ (axis 1) and 0.39 counts $s^{-1}$ $mW^{-1}$ $ppbv^{-1}$ (axis 2). The gated sensitivities are 0.833 counts $s^{-1}$ $mW^{-1}$ $ppbv^{-1}$ (axis 1) and 0.24

counts $s^{-1}$ $mW^{-1}$ $ppbv^{-1}$ (axis 2). Sensitivity differences from COFFEE are due to the combined effects of multiple factors: the changes in optical filters used, differences in individual PMT sensitivities, and widening the time gate. A discussion of ISAF sensitivities is included in St. Clair et al., 2017.



### 3.2 Precision and measurement uncertainty

The main source of noise for CAFE HCHO is Raman and Raleigh scattering of the laser beam. Chamber scatter also significantly contributes to the noise and varies in magnitude with the alignment of the laser through the optical cell. Figure 6 shows the standard deviation of HCHO as a function of HCHO mixing ratio for data processed using exemplar fitting (red symbols) and gated counts (blue symbols) from multiple laboratory experiments. At [HCHO] = 0 ppbv, the average standard deviation for the exemplar-fitted data was ±152 pptv for 1 s and ±87 pptv for 10 s data. For gated count data, the standard deviation was lower: ±100 pptv for 1 s and ±37 pptv for 10 s data. These values are somewhat higher than reported in St. Clair et al., 2017 for COFFEE despite CAFE's higher laser power due to the changes in optical filters and the increased delay from the laser pulse to the start of the fit window in CAFE data processing. The precision can be improved by shortening the delay between the laser pulse and the start of the fit window, thereby increasing the amount of fluorescence data used in the fit. With the $H_2O$ Raman signal effectively blocked, shortening the delay may now be appropriate but further investigation is necessary. For UT/LS measurements of HCHO, the requisite precision exceeds what is possible using exemplar fitting with the current fluorescence filters. Blocking signal from $H_2O$ Raman, however, is not necessary at the low specific humidity of the UT/LS and the water-blocking long pass filters will not be used for high altitude projects, thereby increasing the fluorescence signal. The gated count data also provides an improvement in precision over the exemplar fits, as shown in Fig. 6, and will be available for higher signal-to-noise measurements in the UT/LS.

Figure 7 shows the Allan-Werle deviation as a function of averaging time for two-parameter fit data with [HCHO] = 6 ppbv to illustrate how precision improves with increased data averaging. HCHO data precision improves with averaging up to ~200 s, after which additional time averaging has no benefit. A linear fit to the data with $\tau \leq 256$ s gives a slope (-0.58) similar to the value expected from white noise (0.5), suggesting that white noise dominates on those time scales. The overall HCHO measurement uncertainty is a combination of the calibration uncertainty (±10%) and the less easily quantified uncertainties from unidentified measurement interferences and fit biases. To account for both the known and unknown uncertainties, the overall measurement uncertainty is conservatively estimated to be ± (20% of [HCHO] + 100 pptv).

### 3.3 Time response

For airborne in situ observations, the effective spatial resolution is limited both by the data acquisition rate and the time response of the instrument. To understand the time response of CAFE, pulses of 1.3 ppmv HCHO gas were introduced into a high flow of ultrahigh purity air by a rapidly switching valve (The Lee Company, IEP series) and CAFE subsampled this flow. The decay of the 10 Hz HCHO signal after the pulses was fit using an exponential function, as shown in Fig. 8, and the measured instrument time response is 158±10 (1-σ) ms. An additional set of measurements was conducted for CAFE with the particle filter in-line with the sample flow to verify that the filter does not adversely affect the instrument time response. With the filter in-line, the e-fold time (161 ms) was nearly identical to the response without the filter.



## 4 Field deployment

### 4.1 Aircraft integration

CAFE has successfully flown on the NASA DC-8, ER-2, and C-23 Sherpa aircraft, the Hanseo University King Air during KORUS-AQ, and the University of Maryland Cessna 402B. The rectangular box shape of the chassis (Fig. S1) and internal
pump make CAFE a simple instrument to integrate. Fasteners attach CAFE to structural angles from below, using mounting holes with blind locking nut plates on the bottom of the instrument. For tropospheric aircraft projects, the angles are riveted to non-pivoting cabinet slides (General Devices) to facilitate instrument access. The slides are mounted to the aircraft rack and the slide assembly is secured to the aircraft rack with custom retaining brackets and fasteners. For high altitude aircraft, the angles mount directly to the aircraft rack— photos of the CAFE installation in the left superpod midbody of the NASA
ER-2 are shown in Fig. S3.

### 4.2 Data Comparison with ISAF

During the boreal fall deployment (2017) of the NASA Atmospheric TOMography (ATom) campaign (https://espo.nasa.gov/atom/content/ATom), CAFE joined ISAF on the NASA DC-8 aircraft and sampled from the same inlet. While most ATom flights were in the remote atmosphere where [HCHO] is low, the first test flight on Sept. 14, 2017
out of Palmdale, CA provided good dynamic range with [HCHO] ranging from 100 pptv to over 3 ppbv as shown in Fig. 9. CAFE was operated for ~3 hrs of the 4.3 hr flight to allow the instrument operator to focus on ISAF, the primary instrument for the campaign. The data in Fig. 9 is plotted as both (a) a time series along with altitude and (b) a cross plot with a linear regression. The time series clearly demonstrates the superior precision of the ISAF measurement. The cross plot and linear regression (York et al., 2004) show a good agreement between the two instruments, with a slope of $1.04 \pm 0.02$ and a y-
intercept of $24 \pm 7$ pptv.

The agreement was improved through the application of an ozone-dependent correction to the CAFE data, particularly for the time period of 22.5-23 hrs where the DC-8 passed through stratospherically influenced air. Earlier ATom circuits revealed a strong positive correlation between $O_3$ and ISAF HCHO in stratospheric air masses, with 0.08 pptv of HCHO
produced from exposure to every 1 ppbv of $O_3$. Laboratory tests revealed that prolonged exposure to $O_3$ in lab eliminated the interference in subsequent flights: no $O_3$ correction was needed for ISAF during ATom-3. CAFE was not conditioned in a similar manner before ATom-3 and a correction of 1.2 pptv of HCHO per 1 ppbv of $O_3$ was necessary for ATom-3 data. Possible sources of the higher HCHO yield in CAFE include newer instrument components and the use of THV fluoropolymer tubing, which is no longer used in CAFE for sampling.



## 5 Summary

Improvements to the NR-LIF technique for HCHO from St. Clair et al., 2017 include a higher power laser, fluorescence filters that effectively block $H_2O$ Raman signal, and a more compact form factor. The new model of instrument, CAFE, has

successfully flown on multiple aircraft platforms and obtained the first NR-LIF airborne intercomparison data with a gold-standard HCHO instrument, ISAF. CAFE currently achieves a $1\sigma$ precision of 152 pptv for 1 s (87 pptv for 10 s) data at 0 pptv HCHO. The compact size and simple operation make CAFE an ideal instrument for research-grade HCHO measurements from high altitude aircraft and smaller aircraft focused on local/regional air quality.

## 7 Data availability

CAFE ATom TF01 data are available at https://doi.org/10.7910/DVN/DGALD0 (St. Clair, 2019). All other ATom data are available at https://doi.org/10.3334/ORNLDAAC/1581 (ATom Science Team, 2017).

**The Supplement related to this article is available online.**

*Acknowledgements.* JSC thanks Glenn Wolfe for insightful comments on the manuscript. Thanks to the National Suborbital
Research Center for the ATom altitude data and the NOAA UCATS team (James Elkins, Fred Moore, Eric Hintsa) for the $O_3$ measurements used to correct CAFE ATom data. Funding was provided by NASA Airborne Instrument Technology Transition (AITT), Upper Atmospheric Research, and Tropospheric Composition programs.

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



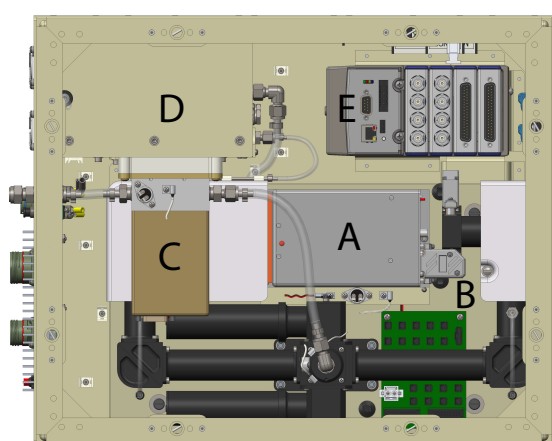

**Figure 1. A top view of the CAFE instrument, with major components labeled: (a) laser, (b) optical plate, (c) pressure controller, (d) pump, and (e) RIO data acquisition system.**

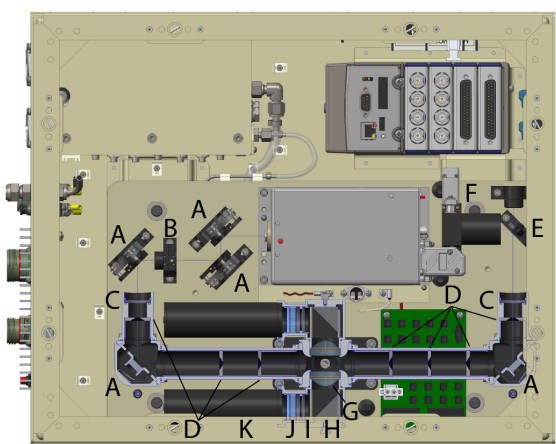

**Figure 2. CAFE instrument top view, with dust covers removed and a cutaway of the detection cell to show its optical components: (a) steering mirrors, (b) half-wave plate and collimating lens, (c) cell windows, (d) laser baffles, (e) beam sampler, (f) laser power monitor, (g) aspheric lens, (h) prism dielectric mirror, (i) optical filters, (j) lens, and (k) photomultiplier tube.**




(a)

(b)

**Figure 3. The combined optical filter transmission spectrum for (a) detection axis 1 and (b) detection axis 2. HCHO fluorescence and $N_2$, $O_2$, and $H_2O$ Raman spectra are shown with arbitrary units. The Raman spectra are for the first Stokes Q branch only and are intended to give a general sense of the transition wavelengths.**





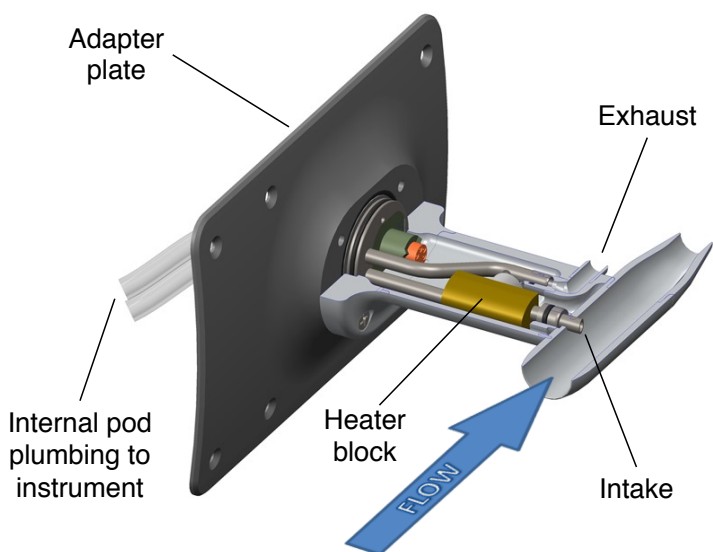

**Figure 4. A cross-sectional view of the CAFE inlet used on the NASA ER-2. Ram pressure is created by ambient air entering a**
5 **nacelle mounted on the end of an air foil. Sample air is drawn toward the instrument through a heated Silcosteel tube that**
**protrudes into the nacelle perpendicular to the ambient air flow. Internal to the pod, 1.3 m of tubing delivers the sample flow to**
**the instrument and an equal amount returns the exhaust flow to the inlet where it is directed out the back of the air foil.**





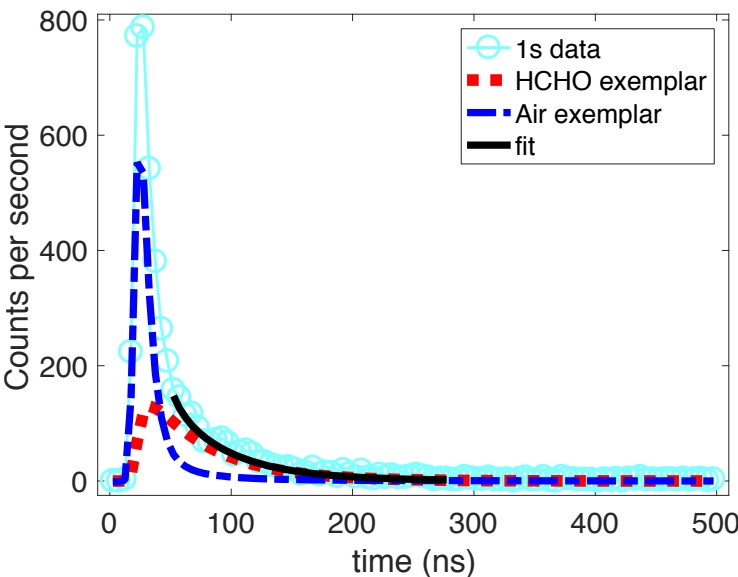

**Figure 5. An example 1 Hz, 5 ns binned time profile (cyan circles) for ~ 10 ppbv HCHO is fit using a two-parameter nonlinear least squares approach. The two components of the fit are the scaled air (dashed blue line) and HCHO (dashed red line) contributions to the time profile. The fit, shown in black, is performed over a subset of the time profile.**

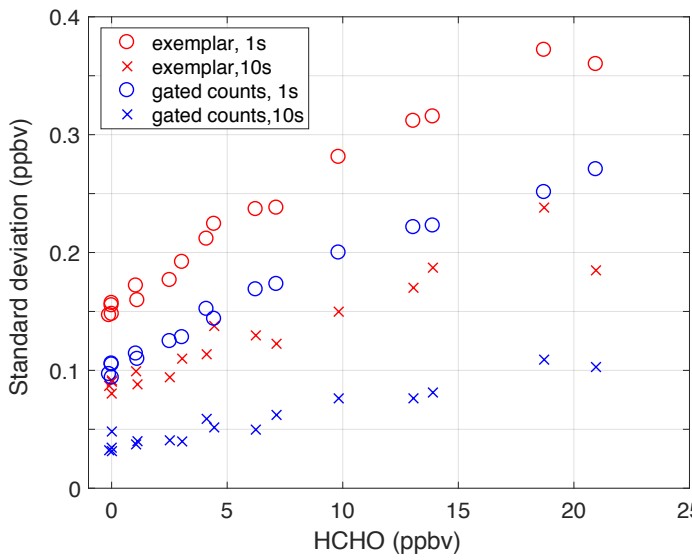

**Figure 6. Standard deviation as a function of HCHO for two data processing techniques: exemplar fitting for 1 s (red circle) and 10 s (red x) data, and gated counts for 1 s (blue circle) and 10 s (blue x) data.**





**Figure 7. Normalized Allan-Werle deviation verses averaging time for [HCHO] = 6 ppbv. HCHO precision improves with averaging up to ~200 s. A linear fit (red dashed line) to the data up to 256 s gives a slope (-0.58) similar to the value expected from white noise (0.5).**

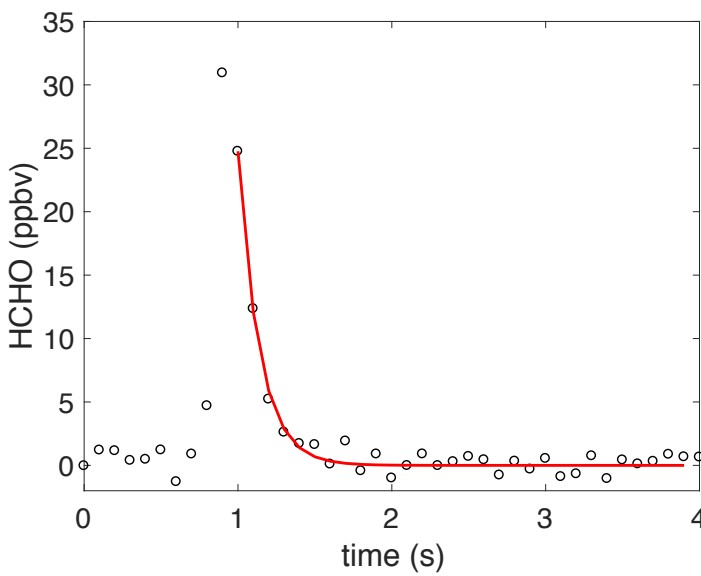

**Figure 8. Pulses of HCHO were fit to provide the e-fold flush time (158±10 ms, 1-σ) of the instrument.**



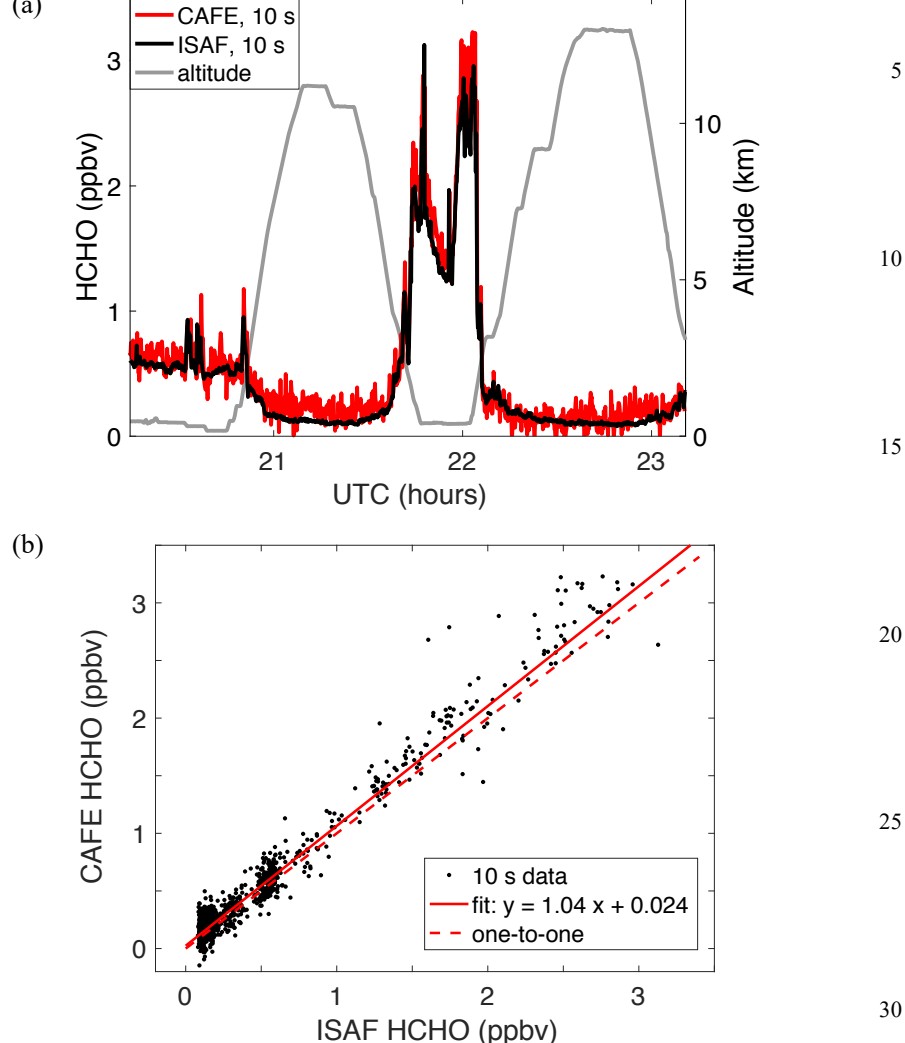

**Figure 9. (a) Time series data from TF01 of ATom-3 show good agreement between CAFE (red) and ISAF (black). (b) A linear regression of the data yields a slope of 1.04 ± 0.02 and intercept of 0.024 ± 0.007. Data from both instruments were averaged to 10 s.**