# Peer review of "CAFE: a new, improved non-resonant laser-induced fluorescence instrument for airborne in situ measurement of formaldehyde"

_Atmospheric Measurement Techniques, 2019_

## Referee Comment (RC1) · Anonymous Referee #2 · 16 Jun 2019

The manuscript by St. Clair et al. describes the NASA Compact Formaldehyde Experiment (CAFE) instrument which is the successor to the NASA Compact Formaldehyde Fluorescence Experiment (COFFEE) instrument. The authors provide a thorough and very detailed description of all instrument components on CAFE and show some intercomparison data from ATom-3 comparing CAFE (a non-resonant laser-induced fluorescence (LIF) instrument) to a resonant LIF instrument (ISAF).

In terms of general comments, there needs to be more discussion or data intercomparison with COFFEE since the manuscript's title emphasizes the idea that CAFE is the new and improved version of COFFEE. CAFE is clearly more versatile than COF-

[Figure]

FEE in terms of flying on different types of aircraft, but when CAFE and COFFEE's precision are compared in Section 3.2, COFFEE is shown to be slightly outperforming CAFE (even with CAFE's higher laser power). Explicitly listing out all of the design differences between CAFE and COFFEE in a single section or table will help the reader to quickly understand how this manuscript advances and improves upon the COFFEE manuscript (St. Clair et al., AMT, 2017).

The manuscript fits well within the scope of AMT, and I recommend publication after attention to the previous general comment and the following specific comments/technical corrections.

Specific Comments:

- Page 3, Line 9: (Minor comment) Just to aid in reader understanding, please explain in the manuscript why the baffle opening diameter increases slightly in size as the baffles approach the detection cell.

- Page 3, Line 13-14: Why isn't the inner surface of the detection cell also coated with Acktar (like the baffles) in order to minimize scatter?

- Section 2: It's clear that the two detection axes have different filters, but there should be a statement somewhere in Section 2 that explicitly mentions the rationale for the two detection axes. It's not immediately clear from the text how the data from the two detection axes are combined or used to get the 1s data (cyan circles) shown in Figure 5.

- Page 7, Line 6: Please state the humidity that this zero was performed at since it seems like CAFE's accuracy is affected by whether there's water present in the air sample, and the instrument's precision is affected by the presence of the water-blocking long pass filters. A similar dry experiment should be done without the water-blocking long pass filters in place and its results mentioned in the manuscript (to represent UT/LS conditions).

- Figure 3: Mention in the caption how much of the possible HCHO fluorescence signal is attenuated by the filters.

Technical Corrections:

- Abstract: Formadehyde should be Formaldehyde

- Abstract: Mention CAFE accuracy in abstract of +/- 20%[HCHO] + 100 ppt

- Page 5, Line 4: Use Celsius

- Page 6, Line 21: sLm should be defined back on Page 4, Line 26 since that's the first appearance of the unit

- Figure 2: Beam path is faint and hard to see in figure

- Figure 9: Mention that the York fit is used in the figure caption
* * *

---

## Referee Comment (RC2) · Anonymous Referee #1 · 18 Jun 2019

St. Clair et al. describe a newly developed non-resonant laser induced fluorescence instrument for measurements of formaldehyde. This team is quite experienced with LIF measurements of formaldehyde and has published descriptions of two other of these instruments, COFFEE and ISAF, which are also discussed in the present manuscript. The manuscript is well written and does an excellent job of describing details of the instrument design and operation. I have no concerns about the scientific quality of the work and believe that the paper should be published in AMT after considering my comments below.

My only general suggestion is that the authors might provide some context or discussion of the data quality relative to measurement requirements for their target sampling regions. For example, CAFE is described as being well suited for high altitude operation, but it is not clear to me that the data quality are sufficient for measuring formaldehyde in the UT/LS region. Figure 9 shows that in the UT regions sampled by the DC-8 there are some quite significant discrepancies between the ISAF measurements and CAFE. From this figure it appears that even with more averaging, at times the difference between ISAF and CAFE might exceed 100%. I presume this is due to the as stated $\pm$ 100 pptv in the CAFE zero. Perhaps the authors could discuss for example what measurement quality is required to understand the role of HCHO in the UT HOx budget or to identify convective events, scientific issues which they mention in the introduction.

Specific points: 1) Page 7, line 14. How much more signal can be gained by excluding the H2O Raman filter and how would this translate to S:N? 2) Figure 9a, extend HCHO axis to show negative values.

---

## Author Comment (AC1) · 29 Jul 2019

The authors thank the two reviewers for their thoughtful comments. A reply to each comment, including changes to the manuscript, is included below with the original comment in *italic* font and the reply in normal font.

**Referee #1:**

My only general suggestion is that the authors might provide some context or discussion of the data quality relative to measurement requirements for their target sampling regions. For example, CAFE is described as being well suited for high altitude operation, but it is not clear to me that the data quality are sufficient for measuring formaldehyde in the UT/LS region. Figure 9 shows that in the UT regions sampled by the DC-8 there are some quite significant discrepancies between the ISAF measurements and CAFE. From this figure it appears that even with more averaging, at times the difference between ISAF and CAFE might exceed 100%. I presume this is due to the as stated  $\pm 100$  pptv in the CAFE zero. Perhaps the authors could discuss for example what measurement quality is required to understand the role of HCHO in the UT HOx budget or to identify convective events, scientific issues which they mention in the introduction.

**Author reply:**

This is an excellent point regarding the measurement demands of a given science objective. For most tropospheric sampling goals, the stated  $\pm 20\% + 100$  pptv uncertainty is sufficient. For UT/LS sampling, the +100 pptv term becomes an issue for science goals where the absolute value of the observed HCHO matters, such as the HOx budget. For convective events, the relative change in HCHO outside/inside a recently convected air parcel will still be scientifically useful in spite of a large fractional uncertainty in the absolute value of the background air measurement. Good visual references for expected UT/LS HCHO from convection are Fig. 6 in Fried et al., 2008 and Fig. 3 in Fried et al., 2016.

For science goals that require higher accuracy, the instrument must be configured to reduce the 100 pptv term in the uncertainty. The instrument can be zeroed in-flight, either by using a scrubber such as 2,4-Dinitrophenylhydrazine (DNPH) or Drierite + molecular sieve, or by flying a zero air cannister.

In order to better contextualize the uncertainty value, the following text was added to Section 3.2:

For specific projects that require higher accuracy, the instrument must be configured to reduce the 100 pptv term in the uncertainty. The instrument can be zeroed in-flight, either by using a scrubber such as 2,4-Dinitrophenylhydrazine (DNPH) or Drierite + molecular sieve, or by flying a zero air cannister. The addition of a zeroing system will likely impact the instrument time response.

**Specific points:**

1) Page 7, line 14. How much more signal can be gained by excluding the H2O Raman filter and how would this translate to S:N?

**Author reply:**

Using an old laboratory experiment with the original fluorescence filters, we can provide precision values for comparison. The following text was added to Section 3.2:

Using the same fitting window and gate delay but with the fluorescence filters from St. Clair et al., 2017, the average standard deviation at [HCHO] = 0 ppbv for a laboratory experiment was  $\pm 106$  pptv for 1 s and  $\pm 62$  pptv for 10 s data (exemplar-fitted) and  $\pm 55$  pptv for 1 s and  $\pm 18$  pptv for 10 s data (gated data).

2) Figure 9a, extend HCHO axis to show negative values.

**Author reply:**

The Figure 9a y-axis limits were adjusted to show all values.

**Referee #2:**

In terms of general comments, there needs to be more discussion or data intercomparison with COFFEE since the manuscript's title emphasizes the idea that CAFE is the new and improved version of COFFEE. CAFE is clearly more versatile than COFFEE in terms of flying on different types of aircraft, but when CAFE and COFFEE's precision are compared in Section 3.2, COFFEE is shown to be slightly outperforming CAFE (even with CAFE's higher laser power). Explicitly listing out all of the design differences between CAFE and COFFEE in a single section or table will help the reader to quickly understand how this manuscript advances and improves upon the COFFEE manuscript (St. Clair et al., AMT, 2017).

**Author reply:**

To highlight the design differences between CAFE and COFFEE, the following section was added to the text:

**2.7 Summary of updates from COFFEE**

CAFE applies the same measurement technique and uses many of the same components as COFFEE, but also contains a number of updates. The updates are summarized here to aid the reader in understanding the differences between the two instruments. CAFE uses a new version of the Explorer One laser (Section 2.1) operated at twice the repetition rate (80 kHz), yielding a 60% increase in average laser power. The fluorescence filters (Section 2.1) were changed to block Raman scattering signal from water vapor that caused a fitting bias in data processing. The additional filters, which decrease the instrument sensitivity to HCHO, are necessary for tropospheric observations but not for UT/LS observations where water vapor is low. The shape of CAFE's chassis (Fig S1) is more compact and easier to integrate onto most aircraft platforms; the unusual shape of COFFEE is due to the available volume in the Alpha Jet pod. The compact CAFE chassis is made possible by including two additional turning mirrors at 90° bends in the baffle arms. A ballast volume (Section 2.1) between the detection cell and the pump was added to muffle pressure fluctuations from the diaphragm pump and prevent them from propagating into the detection cell. Heaters were added to the pump (Section 2.4) to

maintain its optimal operation temperature during high altitude flights.

**Specific Comments:**

- Page 3, Line 9: (Minor comment) Just to aid in reader understanding, please explain in the manuscript why the baffle opening diameter increases slightly in size as the baffles approach the detection cell.

**Author reply:**

The text was changed to read:

Baffle opening diameter increases from entrance to exit over the 7 baffles with three sizes (2.5 mm, 3 mm, 3.5 mm) to accommodate any beam divergence.

- Page 3, Line 13-14: Why isn't the inner surface of the detection cell also coated with Acktar (like the baffles) in order to minimize scatter?

**Author reply:**

Good question. The low scatter coatings have a high surface area and consequently would have an adverse effect on the time response of the measurement if applied to the detection cell walls. The baffles are downstream of the detection cell and do not impact the instrument time response. Also, reducing chamber scatter would not benefit the S/N as much as you might expect because over half of the background counts are from Raleigh scatter by  $N_2$  and  $O_2$ .

- Section 2: It's clear that the two detection axes have different filters, but there should be a statement somewhere in Section 2 that explicitly mentions the rationale for the two detection axes. It's not immediately clear from the text how the data from the two detection axes are combined or used to get the 1s data (cyan circles) shown in Figure 5.

**Author reply:**

To address the reviewer's comment, text was added in Section 2.2 (Optical system) and Section 2.6 (data processing). The 1s data (cyan circles) is shown for Axis 2 only—all the fitting is done separately for each axis and data from the two axes are averaged together once they are in mixing ratio form.

Section 2.2, end of section:

Axis 2, with the 11-band band pass filter, is  $\sim$ 3x less sensitive than Axis 1. The 11-band filter, however, provides greater confidence in the non-resonant LIF measurement selectivity. With more ambient observations, the Axis 1 filters may prove to be sufficiently selective to be used on both axes.

**Section 2.6, middle of first paragraph:**

After the long-lived signal is removed, the 0-500 ns time profile is fit using a twoparameter nonlinear least squares optimization as shown in Fig. 5 for Axis 2.

Section 2.6, end of first paragraph:

Mixing ratio data from the two axes are averaged together to provide the final data product.

- Page 7, Line 6: Please state the humidity that this zero was performed at since it seems like CAFE's accuracy is affected by whether there's water present in the air sample, and the instrument's precision is affected by the presence of the water-blocking long pass filters. A similar dry experiment should be done without the water-blocking long pass filters in place and its results mentioned in the manuscript (to represent UT/LS conditions).

**Author reply:**

With the new optical filters, CAFE's accuracy is unaffected by the presence of water. To clarify the very low humidity conditions of the data used to generate the precision numbers, the following line was modified to specify the gas used:

Figure 6 shows the standard deviation of HCHO as a function of HCHO mixing ratio for data processed using exemplar fitting (red symbols) and gated counts (blue symbols) from multiple laboratory experiments using UHP air.

Using an old laboratory experiment with the original fluorescence filters, we can provide precision values for comparison. The following text was added to Section 3.2:

Using the same fitting window and gate delay but with the fluorescence filters from St. Clair et al., 2017, the average standard deviation at [HCHO] = 0 ppbv for a laboratory experiment was  $\pm 106$  pptv for 1 s and  $\pm 62$  pptv for 10 s data (exemplar-fitted) and  $\pm 55$  pptv for 1 s and  $\pm 18$  pptv for 10 s data (gated data).

- Figure 3: Mention in the caption how much of the possible HCHO fluorescence signal is attenuated by the filters.

**Author reply:**

To address this comment, we approximated the amount of the fluorescence transmitted by the filters using the data shown in Figure 3 (published HCHO fluorescence and manufacturer-provided filter transmission data). The Figure 3 caption now reads:

Figure 3. The combined optical filter transmission spectrum for (a) detection Axis 1 and (b) detection Axis 2. HCHO fluorescence and  $N_2$ ,  $O_2$ , and  $H_2O$  Raman spectra are shown with arbitrary units. The Raman spectra are for the first Stokes Q branch only and are intended to give a general sense of the transition wavelengths. Filters transmit approximately 34% and 19% of the fluorescence for Axis 1 and Axis 2, respectively.

Technical Corrections:

- Abstract: Formadehyde should be Formaldehyde

Author reply: Fixed.

- Abstract: Mention CAFE accuracy in abstract of +/- 20%[HCHO] + 100 ppt

**Author reply:**

The following line was added to the Abstract:

The accuracy of CAFE, using its curve-fitting data processing, is estimated as  $\pm 20\%$  of [HCHO] + 100 pptv.

- Page 5, Line 4: Use Celsius

Author reply: Changed to Celsius.

- Page 6, Line 21: sLm should be defined back on Page 4, Line 26 since that's the first appearance of the unit

**Author reply:** sLm definition moved to page 4, line 26.

- Figure 2: Beam path is faint and hard to see in figure

Author reply: The beam path line was thickened and the color changed to be more visible.

- Figure 9: Mention that the York fit is used in the figure caption

**Author reply:** A citation for the York fit was added to the Figure 9 caption.

**References**

St. Clair, J. M., Swanson, A. K., Bailey, S. A., Wolfe, G. M., Marrero, J. E., Iraci, L. T., Hagopian, J. G. and Hanisco, T. F.: A new non-resonant laser-induced fluorescence instrument for the airborne in situ measurement of formaldehyde, Atmos. Meas. Tech., 10(12), 4833–4844, doi:10.5194/amt-10-4833-2017, 2017.

Fried, A., Olson, J. R., Walega, J. G., Crawford, J. H., Chen, G., Weibring, P., Richter, D., Roller, C., Tittel, F., Porter, M., Fuelberg, H., Halland, J., Bertram, T. H., Cohen, R. C., Pickering, K.,

Heikes, B. G., Snow, J. A., Shen, H., O'Sullivan, D. W., Brune, W. H., Ren, X., Blake, D. R.,
Blake, N., Sachse, G., Diskin, G. S., Podolske, J., Vay, S. A., Shetter, R. E., Hall, S. R.,
Anderson, B. E., Thornhill, L., Clarke, A. D., McNaughton, C. S., Singh, H. B., Avery, M. A.,
Huey, G., Kim, S. and Millet, D. B.: Role of convection in redistributing formaldehyde to the
upper troposhere over Noth Atlantic during the summer 2004 INTEX campaign, J. Geophys. Res.
Atmos., 113(17), 1–19, doi:10.1029/2007JD009760, 2008.
Fried, A., Barth, M. C., Bela, M., Weibring, P., Richter, D., Walega, J., Li, Y., Pickering, K.,
Apel, E., Hornbrook, R., Hills, A., Riemer, D. D., Blake, N., Blake, D. R., Schroeder, J. R., Luo,
Z. J., Crawford, J. H., Olson, J., Rutledge, S., Betten, D., Biggerstaff, M. I., Diskin, G. S., Sachse,

G., Campos, T., Flocke, F., Weinheimer, A., Cantrell, C., Pollack, I., Peischl, J., Froyd, K., Wisthaler, A., Mikoviny, T. and Woods, S.: Convective transport of formaldehyde to the upper troposphere and lower stratosphere and associated scavenging in thunderstorms over the central United States during the 2012 DC3 study, J. Geophys. Res. Atmos., 121(12), 7430–7460, doi:10.1002/2015JD024477, 2016.